# Trends in degenerative lumbar spinal surgery during the early COVID-19 pandemic in Republic of Korea: A national study utilizing the national health insurance database

Woon Tak Yuh[1,2], Jinhee Kim[3], Mi-Sook Kim[3,4‡], Jun-Hoe Kim[5], Young Rak Kim[5], Sum Kim[5], Chun Kee Chung[5,6,7], Chang-Hyun Lee[5,6], Sung Bae Park[5,6,8], Kyoung-Tae Kim[9,10], John M. Rhee[11], Young San Ko[9], Chi Heon Kim[5,6,12‡*]

1 Department of Neurosurgery, Hallym University College of Medicine, Chuncheon-si, Gangwon-do, Republic of Korea, 2 Department of Neurosurgery, Hallym University Dongtan Sacred Heart Hospital, Hwaseong-si, Gyeonggi-do, Republic of Korea, 3 Division of Clinical Epidemiology, Medical Research Collaborating Center, Seoul National University Hospital, Seoul, Republic of Korea, 4 Department of Preventive Medicine, Seoul National University College of Medicine, Seoul, Republic of Korea, 5 Department of Neurosurgery, Seoul National University Hospital, Seoul, Republic of Korea, 6 Department of Neurosurgery, Seoul National University College of Medicine, Seoul, Republic of Korea, 7 Department of Brain and Cognitive Sciences, Seoul National University, Seoul, Republic of Korea, 8 Department of Neurosurgery, Seoul National University Boramae Hospital, Boramae Medical Center, Seoul, Republic of Korea, 9 Department of Neurosurgery, School of Medicine, Kyungpook National University, Daegu, Republic of Korea, 10 Department of Neurosurgery, Korea University Guro Hospital, Korea University College of Medicine, Republic of Korea, 11 Department of Orthopaedic Surgery, Emory University School of Medicine, Atlanta, Georgia, United States of America, 12 Department of Medical Device Development, Seoul National University College of Medicine, Seoul, Republic of Korea

☯ These authors contributed equally to this work.
‡ M-SK and CHK contributed equally to this work as co-corresponding authors
* chiheon1@snu.ac.kr

## Abstract

During the first year of the COVID-19 pandemic, the Republic of Korea (ROK) experienced three epidemic waves in February, August, and November 2020. These waves, combined with the overarching pandemic, significantly influenced trends in spinal surgery. This study aimed to investigate the trends in degenerative lumbar spinal surgery in ROK during the early COVID-19 pandemic, especially in relation to specific epidemic waves. Using the National Health Information Database in ROK, we identified all patients who underwent surgery for degenerative lumbar spinal diseases between January 1, 2019 and December 31, 2020. A joinpoint regression was used to assess temporal trends in spinal surgeries over the first year of the COVID-19 pandemic. The number of surgeries decreased following the first and second epidemic waves (p<0.01 and p = 0.34, respectively), but these were offset by compensatory increases later on (p<0.01 and p = 0.05, respectively). However, the third epidemic wave did not lead to a decrease in surgical volume, and the total number of surgeries remained comparable to the period before the pandemic. When compared to the pre-COVID-19 period, average LOH was reduced by 1 day during the COVID-19 period (p<0.01), while mean hospital costs increased significantly from 3,511 to 4,061 USD (p<0.01). Additionally, the transfer rate and the 30-day readmission rate significantly

**Data Availability Statement:** In the Republic of Korea (ROK), nearly all residents are beneficiaries of the national health insurance system (NHIS),

and all nationwide inpatient and outpatient data regarding diseases and services (i.e., procedures and operations) are coded and registered in the National Health Information Database (NHID) and the Health Insurance Review & Assessment Service (HIRA) database, thus enabling population-based studies to be conducted. The NHID contains demographic characteristics and insurance eligibility information, as well as data on almost all hospitalizations and outpatient medical services. This information encompasses diagnoses, which are based on International Classification of Diseases 10th Revision codes, procedures, prescriptions, examinations, and direct medical costs. We assert that there were no special privileges accorded to the authors in accessing the NHID data. The process followed adheres strictly to the standard protocol provided by NHIS, ensuring equal opportunity for data access to all researchers. The NHIS-NHID data is available to any researchers who are interested by following the appropriate request and review process outlined on the NHIS website (https://nhiss.nhis.or.kr/bd/ab/bdaba02eng.do). There is a cost for the assess to the NHID and the use of virtual terminal.

**Funding:** This study was supported by Seoul National University Hospital Research Fund (Grant No. 3020230120), Doosan Yonkang Foundation (Grant No. 800-20210527), and Armed Forces Capital Hospital (Grant No. 2023MDD0075), received by Chi Heon Kim. The funders had no role in study design, data collection and analysis, decision to publish, or preparation of the manuscript. There was no additional external funding received for this study.

**Competing interests:** The authors have declared that no competing interests exist.

decreased (both p<0.01), while the reoperation rate remained stable (p = 0.36). Despite the impact of epidemic waves on monthly surgery numbers, a subsequent compensatory increase was observed, indicating that surgical care has adapted to the challenges of the pandemic. This adaptability, along with the stable total number of operations, highlights the potential for healthcare systems to continue elective spine surgery during public health crises with strategic resource allocation and patient triage. Policies should ensure that surgeries for degenerative spinal diseases, particularly those not requiring urgent care but crucial for patient quality of life, are not unnecessarily halted.

## Introduction

The first COVID-19 patient was reported in November 2019 in China, and the World Health Organization officially declared COVID-19 to be a global pandemic on March 11, 2020. In the Republic of Korea (ROK), the first case of COVID-19 was confirmed on January 20, 2020, and the level of quarantine was raised to the highest level on February 23, 2020. The entire nation was locked down, and only essential activities were allowed [1].

The COVID-19 pandemic has presented unprecedented challenges for patients and the healthcare community as a whole [2–11]. Due to the increased number of infected individuals, sick patients, and social distancing, medical resources such as hospital beds and medical personnel have had to be practically re-distributed. While the main guidelines and strategies in every country were similar, the healthcare infrastructure and pandemic responses, including spinal surgery varied among countries [12,13]. For example, in ROK, all people are beneficiary of national health insurance or medical aid, and the access to any hospital is not limited to place. During the COVID-19 pandemic, the government of ROK did not restrict elective surgical procedures officially. On the other hand, in the USA, the COVID-19 pandemic led to a nationwide suspension of elective surgery, including spinal procedures, from March 2020 to May 2020 [14].

Although the government of ROK did not officially restrict elective spinal surgeries during the pandemic, the conditions for such surgeries changed significantly. First, the number of hospital beds available for elective surgery decreased. Although restrictions or delays of elective surgery were not recommended, public hospitals, university hospitals, and large private hospitals were required to secure hospital beds and intensive care unit space for COVID-19 patients to accommodate the increasing number of COVID-19 patients. Additionally, social distancing measures reduced the density of hospital beds in a single hospital room compared to the previous circumstances where there had been up to 6 beds in one room. Second, patients could only be admitted if they had a negative polymerase chain reaction (PCR) test and showed no symptoms of COVID-19. Third, patients were concerned about the possibility of nosocomial COVID-19 infections during their hospital stay [15]. Due to those and other unmentioned reasons, the patterns of medical services in spinal surgery may have been affected.

Surgical interventions for degenerative lumbar spinal disease are typically elective and focused on enhancing quality of life rather than addressing emergencies that threaten life immediately. As a result, these conditions are often assigned a lower priority compared to other spinal diseases such as cervical or thoracic lesions with neurological deficits or spinal traumas requiring urgent surgical interventions. In the initial stages of the COVID-19 pandemic, elective surgeries for degenerative lumbar diseases were frequently deferred or canceled as medical priorities shifted to pandemic management. This practice, although intended to

conserve medical resources for acute pandemic care, was found to potentially worsen the long-term quality of life of these patients, increasing their vulnerability as the pandemic protracted [12,14–19]. Based on this experience, this study was designed to observe the impact of the COVID-19 pandemic on surgeries for patients with degenerative lumbar spinal disease who are vulnerable during pandemics. The primary aim of this study was to examine trends in lumbar spinal surgery during the early COVID-19 pandemic in ROK, according to specific epidemic waves. The secondary aims included comparing detailed outcomes, including length of hospital stay (LOH), hospital costs, discharge disposition, the 30-day readmission rate, and the reoperation rate between before and during the COVID-19 pandemic.

## Materials and methods

### National health information database (NHID)

This study utilized the National Health Information Database (NHID) from the National Health Insurance Service (NHIS) of ROK. In ROK, nearly all residents are beneficiaries of the national health insurance system, and all nationwide inpatient and outpatient data regarding diseases and services (i.e., procedures and operations) are coded and registered in the NHID and the Health Insurance Review & Assessment Service (HIRA) database, thus enabling population-based studies to be conducted [20–22]. The NHID contains demographic characteristics and insurance eligibility information, as well as data on almost all hospitalizations and outpatient medical services. This information encompasses diagnoses, which are based on International Classification of Diseases 10th Revision codes, procedures, prescriptions, examinations, and direct medical costs. Since medical service providers electronically submit claims data to the HIRA for reimbursement, and the HIRA reviews them to determine payment eligibility, the information on procedures and medical costs is highly accurate. Given that almost all hospitals in ROK adhere to the NHIS for reimbursement, healthcare utilization records of almost all patients who underwent lumbar spinal surgery could be extracted from the claims data through the NHID. The data are provided after pseudonymization and can only be accessed remotely in a strictly controlled analysis room. Therefore, the Institutional Review Board waived informed consent. The NHID was accessed from November 8, 2021, through August 11, 2023, for the purposes of this research. During and after data collection, the authors did not have access to information that could identify individual participants. All methods were performed in accordance with the relevant guidelines and regulation, and the Institutional Review Board of the Seoul National University Hospital approved the review and analysis of the data (2012-014-1177).

### COVID-19 epidemic waves in ROK

While there is no universally accepted definition of an epidemic wave, Salyer et al. defined an epidemic wave or phase as a rising number of COVID-19 cases with a defined peak, followed by a decline in cases or trough period, in which transmission had decreased [23]. We retrieved the data for daily confirmed COVID-19 cases in 2020 from the Korea Disease Control and Prevention Agency (KDCA) website (https://ncov.kdca.go.kr/en/) with an aim to investigate patterns and determine if there was any epidemic wave. Using the definition provided by Salyer et al., we identified three distinct epidemic waves in ROK during the initial year of the COVID-19 pandemic. These occurred from February 18th to March 14th, August 13th to September 5th, and November 6th to December 31st, 2020. Each wave reached its peak on February 29th, August 27th, and December 25th, 2020, respectively (Fig 1A).

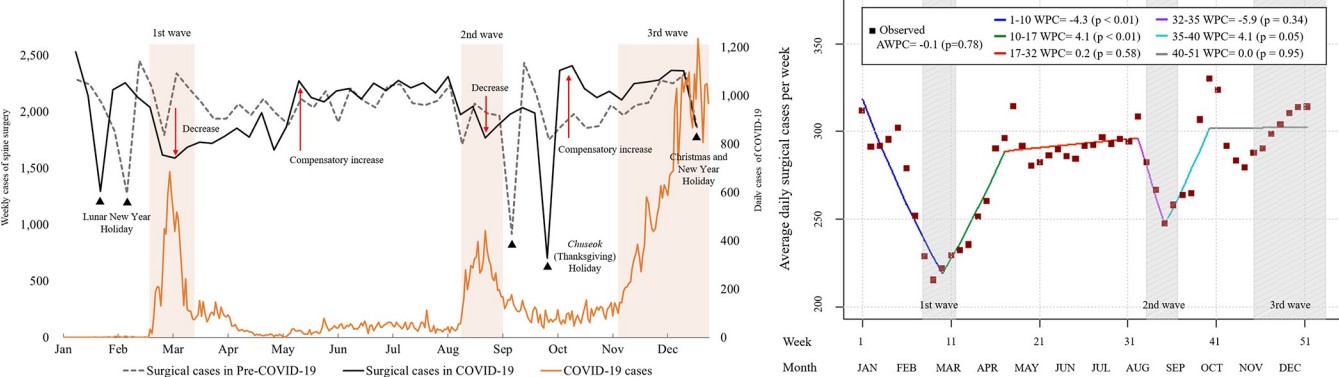

**Fig 1. Analysis of surgical operations in relation to the epidemic waves in ROK.** (a) Weekly count of operations juxtaposed with daily cases of COVID-19. Decreases in weekly operations were observed following the first and second epidemic waves, with compensatory increases occurring in May and October. Notably, the third wave did not result in any discernible decrease or subsequent compensatory increase. (b) Joinpoint regression analysis: A significant decrease in the number of operations was observed during the first epidemic wave ($p<0.01$). The decline during the second epidemic wave did not achieve statistical significance ($p = 0.34$). Following both the first and second waves, compensatory increases in surgical operations were documented (both $p\leq0.05$). Interestingly, neither a decrease nor a compensatory increase was evident during the third epidemic wave. *WPC*, weekly percentage change; *AWPC*, average weekly percentage change.

## Patient inclusion

Patients who underwent lumbar spinal surgery with the diagnosis of herniated intervertebral disc disease (HIVD), lumbar spinal stenosis (LSS), spondylolisthesis, and spondylolysis from 2019 to 2020 were selected to investigate trends in spinal surgery using the number of surgical operations per week. These diagnoses were based on the International Classification of Diseases 10th Revision (ICD-10) codes: HIVD (M51.x), LSS (M48.05, M48.06, M48.07, M48.08), spondylolisthesis (M43.15, M43.16, M43.17, M43.18, M53.26), and spondylolysis (M43.06, M43.07, M43.08). When patients had multiple spinal diseases, the representative disease code followed the hierarchical coding algorithm proposed by Martin et al. [24]. For example, if patients were diagnosed with both spondylolisthesis and HIVD, they were categorized as having spondylolisthesis. Surgical methods were primarily divided into decompression only or fusion surgery. For patients with HIVD, decompression surgery was further divided into open discectomy and endoscopic discectomy.

To compare the periods before and during the COVID-19 pandemic, the patients were divided based on the date of lumbar spinal surgery. Patients who underwent surgery from January 1, 2019 to December 31, 2019 were defined as the pre-COVID-19 cohort, and patients from January 1, 2020 to December 31, 2020 as the COVID-19 cohort. The following exclusion criteria were then applied; 1) a history of spinal fracture, infection, cancer, or inflammatory arthritis; 2) previous lumbar spinal surgery; 3) missing data for sex and year of birth; 4) severe disability (class 1 or 2, ambulation was dependent on aids or a wheelchair or the patient was bedridden); and 5) under age 20 years at surgery. Finally, 78,402 patients in the pre-COVID-19 cohort and 80,165 patients in the COVID-19 cohort were included (Table 1 and Fig 2).

## Variables

**Age and comorbidities.** Patients were categorized by age as young adults (20–39 years), middle-aged adults (40–59 years), old adults (60–69 years), seniors (70–79 years), and octogenarians or more (80–99 years). The Charlson comorbidity index (CCI) was used to assess comorbid conditions [25], and Parkinson's disease and osteoporosis without fracture were also evaluated as other comorbidities. Comorbidities were defined on the basis of

**Table 1. Baseline characteristics.**

|  | Pre-COVID-19 (n = 78,402) | COVID-19 (n = 80,165) | p-value |
|---|---|---|---|
| **Age** |  |  | 0.3235 |
| Mean ± SD | 59.86 ± 14.01 | 59.92 ± 13.99 |  |
| Median [Q1, Q3] | 62 [51, 70] | 62 [51, 70] |  |
| **Age group**, n (%) |  |  | < .0001 |
| 20–39 | 7780 (9.92) | 7838 (9.78) |  |
| 40–59 | 26741 (34.11) | 26729 (33.34) |  |
| 60–69 | 22263 (28.40) | 23918 (29.84) |  |
| 70–79 | 17356 (22.14) | 17359 (21.65) |  |
| 80+ | 4262 (5.44) | 4321 (5.39) |  |
| **Sex**, n (%) |  |  | 0.668 |
| Male | 41172 (52.51) | 42184 (52.62) |  |
| Female | 37230 (47.49) | 37981 (47.38) |  |
| **Charlson Comorbidity index (CCI)** | -0.0249 | < .0001 |  |
| Mean ± SD | 1.29 ± 1.42 | 1.25 ± 1.41 |  |
| Median [Q1, Q3] | 1 [0, 2] | 1 [0, 2] |  |
| **Type of insurance**, n (%) |  |  | 0.005 |
| National Health Insurance | 75350 (96.11) | 77258 (96.37) |  |
| Medical Aid | 3052 (3.89) | 2907 (3.63) |  |
| **Other comorbidities**, n (%) |  |  |  |
| Parkinson's disease | 1111 (1.42) | 1095 (1.37) | 0.3849 |
| Osteoporosis without fracture | 24808 (31.64) | 24778 (30.91) | 0.0016 |
| **Income level**, n (%) |  |  | 0.0030 |
| High (upper 20%) | 26363 (33.63) | 26615 (33.20) |  |
| Middle | 35398 (45.15) | 36851 (45.97) |  |
| Low (lower 20%) | 12321 (15.72) | 12535 (15.64) |  |
| National basic livelihood aid | 3052 (3.89) | 2907 (3.63) |  |
| Missing | 1268 (1.62) | 1257 (1.57) |  |
| **Type of healthcare center**, n (%) |  |  | 0.0030 |
| Tertiary hospital | 8114 (10.35) | 7890 (9.84) |  |
| General hospital | 13040 (16.63) | 13655 (17.03) |  |
| Primary Hospital | 56072 (71.52) | 57400 (71.60) |  |
| Clinic | 1176 (1.50) | 1220 (1.52) |  |
| **Type of disease**, n (%) |  |  | < .0001 |
| Spondylolysis | 1135 (1.45) | 1024 (1.28) |  |
| Spondylolisthesis | 13682 (17.45) | 14027 (17.50) |  |
| LSS | 50925 (64.95) | 53625 (66.89) |  |
| HIVD | 12660 (16.15) | 11489 (14.33) |  |
| **The month of surgery**, n (%) |  |  | < .0001 |
| Jan | 7318 (9.33) | 6915 (8.63) |  |
| Feb | 5750 (7.33) | 6076 (7.58) |  |
| Mar | 6444 (8.22) | 5785 (7.22) |  |
| Apr | 6649 (8.48) | 5868 (7.32) |  |
| May | 6692 (8.54) | 6550 (8.17) |  |
| Jun | 6212 (7.92) | 7245 (9.04) |  |
| Jul | 7400 (9.44) | 7479 (9.33) |  |
| Aug | 6422 (8.19) | 6701 (8.36) |  |

*(Continued)*

**Table 1.** (Continued)

|  | Pre-COVID-19 (n = 78,402) | COVID-19 (n = 80,165) | p-value |
|---|---|---|---|
| Sep | 6116 (7.80) | 5955 (7.43) | |
| Oct | 6155 (7.85) | 6885 (8.59) | |
| Nov | 6184 (7.89) | 7200 (8.98) | |
| Dec | 7060 (9.00) | 7506 (9.36) | |
| **Type of surgery**, n (%) | | | 0.8416 |
| Fusion surgery | 20094 (25.63) | 20581 (25.67) | |
| Decompression | 58308 (74.37) | 59584 (74.33) | |

SD, standard deviation; LSS, lumbar spinal stenosis; HIVD, herniated intervertebral disc disease

corresponding diagnoses on at least one inpatient admission or two outpatient visits within the past 2 years [26–29].

**Types of insurance, income, and hospitals.** The types of insurance were divided into National Health Insurance and Medical Aid. The levels of income were divided into the top 20%, middle, the bottom 20%, and Medical Aid. The hospitals where surgery was performed were categorized based on their size and capacity in ROK, as tertiary referral hospitals ($\geq$ 300 beds), general hospitals (100–300 beds), primary hospitals (30–99 beds), and clinics (30 beds) [21,22,26–29].

**Length of hospital stay and hospital costs.** In ROK, patients usually stay in the hospital for 5–7 days after spinal surgery, but the LOH is not standardized or regulated by the

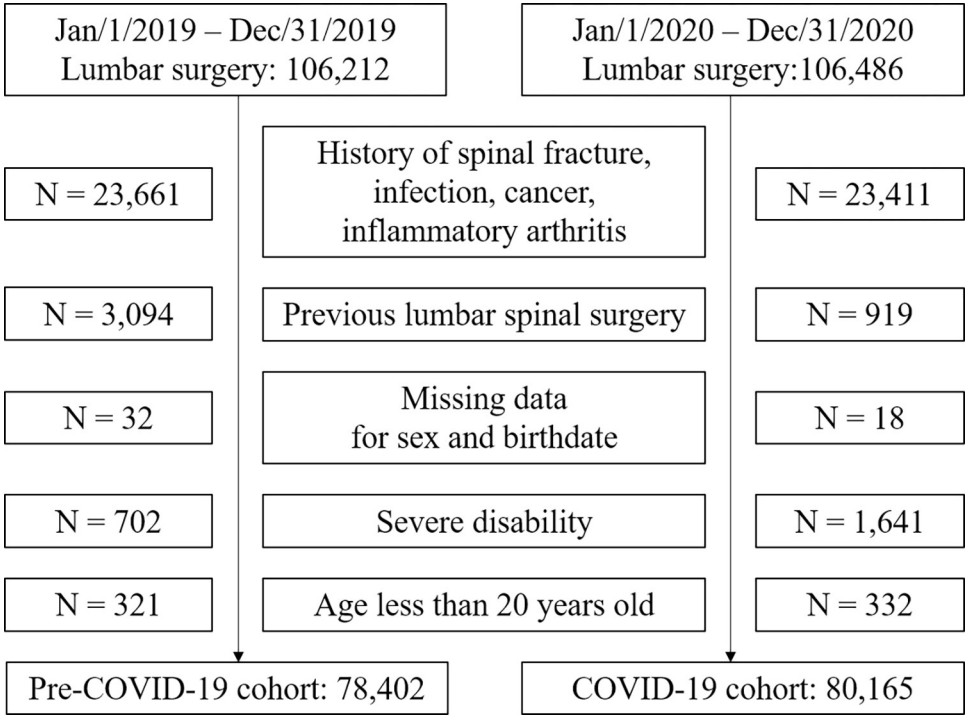

**Fig 2. Flowchart of patients.** The number of patients corresponding to each inclusion and exclusion criteria are noted.

insurance authorities [30]. To determine whether LOH was associated with the readmission rate, patients were subdivided into two groups ($\leq 7$ days and $\geq 8$ days).

The hospital costs for surgery were evaluated, encompassing all expenses incurred during admission, which included cost of surgery, inpatient hospitalization, and surgical implants. The costs were converted based on the 2020 Korean Consumer Price Index (Korean Statistical Information Service Consumer Price Index. [accessed on 09 MAR 2023]; https://kosis.kr/statHtml/statHtml.do?orgId=101&tblId=DT_1J20003&conn_path=I2) and a USD/KRW exchange rate of 1086.3 KRW/USD (Statistics Korea, [accessed on 09 MAR 2023]; https://www.index.go.kr/unity/potal/main/EachDtlPageDetail.do?idx_cd=1068). The proportions of patients were presented on a monthly basis, along with the number of COVID-19 cases. Monthly data for COVID-19 patients were obtained from the Korea Disease Control and Prevention Agency headquarters (https://ncov.kdca.go.kr/).

**Discharge disposition.** Discharge disposition refers to the patient's status upon leaving the hospital, with options including either discharge to home or transfer to another medical institution. Transfer was defined as admission to another medical institution within 2 days of discharge, as direct transfer was not always possible, especially during weekends. Furthermore, the types of medical institutions to which patients were readmitted or transferred after discharge were subdivided into the following: convalescent hospitals, clinics, primary hospitals, general hospitals, tertiary referral hospitals, and oriental medicine hospitals. In ROK, convalescent hospitals function as nursing facilities.

**Thirty-day readmission and reoperation.** Thirty-day readmission was defined as being admitted to the same or a different medical institution within 30 days after discharge. Reoperation was defined as spine-related surgery with a diagnosis of lumbar spinal disease within 30days after the index surgery. Since the NHID does not specify the level of surgery, reoperation included surgery at either the index level or another level.

## Statistical analysis

The baseline characteristics of the pre-COVID-19 and COVID-19 cohorts were presented using the mean and standard deviation, and the median and quartiles, for continuous variables. Categorical variables were presented as the number of patients and proportion. For comparisons between the cohorts, we used the chi-square test for categorical variables and the Wilcoxon rank sum test for continuous variables.

Joinpoint regression analysis was applied to examine changes in time trends of surgical cases over the first year of the COVID-19 pandemic period. This statistical method identifies the time points, called "joinpoints", where linear time trend significantly change and calculated the percentage change between trend-change points, along with the average percentage change in the entire period. The number of spinal surgeries was counted on a weekly basis from January to December 2020, and a 3-week moving average was calculated. To account for the decrease in surgical cases due to weekends or public holidays, such as Lunar New Year and Thanksgiving, the weekly number of surgical cases was adjusted based on the number of working days in those weeks. A permutation test with Monte Carlo resampling was used to determine the optimal number and location of joinpoints. The average weekly percentage change (AWPC) in the entire period was estimated as the weighted average of the estimated weekly percentage change (WPC) of each segment by utilizing the segment lengths as weights. A joinpoint regression plot, with lines connecting consecutive joinpoints, depicted the trends and shifts in data segments over the observed time period.

A logistic regression model was used to compare the rates of reoperation and readmission within 30 days after surgery between pre-COVID-19 and COVID-19 periods. For multivariate

analysis, adjustments were made for age, sex, CCI, income level, length of hospital stay, Parkinson's disease, osteoporosis, type of insurance, and type of hospital. Crude and adjusted odds ratios (ORs), along with their 95% confidence intervals, were estimated. Joinpoint regression analysis was conducted using Joinpoint Regression Program, Version 4.9.1.0 (Statistical Research and Applications Branch; National Cancer Institute, Rockville, MD, USA). The remaining statistical analyses were performed with SAS Enterprise Guide 7.1 (SAS Institute Inc, Cary, NC, USA) and R 4.2.0 (R Foundation for Statistical Computing, Vienna, Austria).

## Results

The overall proportion of fusion surgery did not change during the COVID-19 pandemic (p = 0.84). However, there was a slight change in the proportion according to the diagnosis. In HIVD, the proportion of patients who underwent fusion surgery decreased (p<0.01), while the proportion of patients who underwent endoscopic discectomy increased (p<0.01). In LSS, the proportion of patients who underwent fusion surgery increased (p<0.01) (Table 2).

### Number of operations and monthly distribution

The total number of operations was 78,420 in the pre-COVID-19 period and 80,165 during the COVID-19 period, indicating a slight increase in surgical procedures during the pandemic (Table 1). The majority of operations took place in primary hospitals, followed by general hospitals, tertiary hospitals, and clinics in both timeframes.

An analysis of the weekly number of operations revealed consistent patterns in both cohorts, showing declines around three major holiday seasons: the Lunar New Year, *Chuseok* (Korean Thanksgiving), and the Christmas-New Year holidays. Given that the Lunar New Year and Chuseok holidays are determined by the lunar calendar, the specific dates of these holidays differ annually (Fig 1A). Apart from these holiday seasons, the monthly distribution of surgeries was largely similar between the pre-COVID-19 and COVID-19 cohorts (Table 1). Notably, during the COVID-19 pandemic, there was a decline in surgeries in early March,

**Table 2. Types of surgery.**

|  | Pre-COVID-19 | COVID-19 | p-value |
|---|---|---|---|
| **All patients** | n = 78,402 | n = 80165 |  |
| Fusion surgery | 20094 (25.63) | 20581 (25.67) | 0.8416 |
| Decompression | 58309 (74.37) | 59584 (74.33) |  |
| **Spondylolysis** | n = 1135 | n = 1024 |  |
| Fusion surgery | 759 (66.87) | 695 (67.87) | 0.6212 |
| Decompression | 376 (33.13) | 329 (32.13) |  |
| **Spondylolisthesis** | n = 13682 | n = 14027 |  |
| Fusion surgery | 9796 (71.60) | 10101 (72.01) | 0.4445 |
| Decompression | 3886 (28.40) | 3926 (27.99) |  |
| **LSS** | n = 50925 | n = 53625 |  |
| Fusion surgery | 5446 (10.69) | 6466 (12.06) | < .0001 |
| Decompression | 45479 (89.31) | 47159 (87.94) |  |
| **HIVD** | n = 12660 | n = 11489 |  |
| Open discectomy | 8490 (67.06) | 7960 (69.28) | < .0001 |
| Endoscopic discectomy | 77 (0.61) | 210 (1.83) | < .0001 |
| Fusion surgery | 4093 (32.33) | 3319 (28.89) | < .0001 |

*LSS*, lumbar spinal stenosis; *HIVD*, herniated intervertebral disc disease.

following the first epidemic wave. This suppressed surgical volume persisted through April, and a compensatory increase in surgeries was observed by May. In August, with the onset of the second epidemic wave, there was a decrease in surgeries, but a compensatory increase occurred more quickly than after the first wave. Despite November witnessing the third and most prominent epidemic wave, there was no significant drop in surgeries or a subsequent compensatory rise. (Fig 1A)

## Joinpoint regression

The five joinpoints model was selected as the best-fit model, and during the first year of COVID-19 pandemic, this model identified six segments: 1–10, 10–17, 17–32, 32–35, 35–40, and 40–51 weeks (Fig 1B). The identified first and fourth joinpoints were matched with the first and second epidemic waves. The joinpoint plot revealed a marked decline in operations up to the first joinpoint with WPC of -4.3 ($p < 0.01$). After the first joinpoint, a compensatory increase trend with WPC 4.1 ($p < 0.01$) was observed, up to the second joinpoint in the 17th week in April. After the second joinpoint, there was a plateau with no significant changes until the third joinpoint, at which point a declining trend emerged, with a WPC of -5.9 until the fourth joinpoint, representing the peak of the second epidemic wave. However, this decline was not statistically significant ($p = 0.34$). After the fourth joinpoint, the trend shifted towards an increase with a WPC of 4.1 ($p < 0.05$) until fifth joinpoint, but then there was no further increase. Remarkably, the period corresponding to the third epidemic wave in November showed neither a decrease nor a compensatory increase (Fig 1B).

## LOH and hospital costs

LOH significantly decreased by 1 day or more in patients with spondylolisthesis and HIVD. The proportion of LOH exceeding 8 days was 69.49% in the pre-COVID-19 cohort, while it decreased to 65.86% in the COVID-19 cohort ($p < 0.01$). The mean hospital cost was significantly higher in the COVID-19 cohort than in the pre-COVID-19 cohort, across all diagnoses (from 3,511 to 4,061 USD, $p < 0.01$) (Table 3).

**Table 3. Length of hospital stay and hospital costs during admission.**

|  | Pre-COVID-19 | | COVID-19 | | p-value |
|---|---|---|---|---|---|
|  | mean ± SD | median [Q1, Q3] | mean ± SD | median [Q1, Q3] |  |
| **Length of hospital stay (Day)** | | | | | |
| All patients | 13 ± 10 | 11 [8, 16] | 12 ± 8 | 11 [7, 15] | < .0001 |
| Spondylolysis | 16 ± 9 | 15 [10, 19] | 15 ± 9 | 14 [10, 19] | 0.217 |
| Spondylolisthesis | 16 ± 13 | 15 [11, 19] | 15 ± 9 | 14 [10, 18] | < .0001 |
| LSS | 12 ± 8 | 10 [7, 15] | 11 ± 8 | 10 [6, 14] | < .0001 |
| HIVD | 14 ± 13 | 12 [8, 17] | 13 ± 10 | 11 [8, 16] | < .0001 |
| **Hospital cost (USD)\*** | | | | | |
| All patients | 3511 ± 2766 | 2687 [1810, 4329] | 4061 ± 2892 | 3148 [2283, 5002] | < .0001 |
| Spondylolysis | 4277 ± 2665 | 3624 [2215, 5955] | 5232 ± 2823 | 4870 [3054, 6824] | < .0001 |
| Spondylolisthesis | 5224 ± 3148 | 4991 [2757, 6900] | 6040 ± 3251 | 5757 [3580, 7577] | < .0001 |
| LSS | 2806 ± 2034 | 2370 [1607, 3341] | 3356 ± 2197 | 2813 [2117, 3857] | < .0001 |
| HIVD | 4424 ± 3651 | 3194 [2140, 5691] | 4830 ± 3793 | 3508 [2508, 5917] | < .0001 |

*SD*, standard deviation; *LSS*, lumbar spinal stenosis; *HIVD*, herniated intervertebral disc disease; *USD*, US dollar.

\* Consumer price index was set at the year 2020 and the conversion of won against dollars was 1086.3.

**Table 4. Type of healthcare centers in transfers and 30-day readmissions.**

| | Pre-COVID-19, Number of patients (%) | | | | COVID-19, Number of patients (%) | | | |
|---|---|---|---|---|---|---|---|---|
| | Total operation (n = 78,402) | LOH = <7 (n = 23921, 30.5%) | LOH > = 8 (n = 54481, 69.5%) | p-value | Total operation (n = 80165) | LOH = <7 (n = 27369, 34.1%) | LOH > = 8 (n = 52796, 65.9%) | p-value |
| **Transfer to other hospitals** | **8822 (11.25)** | **1579 (6.60)** | **7243 (13.29)** | **< .0001** | **6554 (8.18)** | **1287 (4.70)** | **5267 (9.98)** | **< .0001** |
| **Type of healthcare center in transfer** | **n = 8822** | **n = 1579** | **n = 7243** | **< .0001** | **n = 6554** | **n = 1287** | **n = 5267** | **< .0001** |
| Convalescent hospital | 1806 (20.47) | 109 (6.90) | 1697 (23.43) | | 1172 (17.88) | 91 (7.07) | 1081 (20.52) | |
| Public health center | 16 (0.18) | 3 (0.19) | 13 (0.18) | | 2 (0.03) | 0 (0.00) | 2 (0.04) | |
| Clinic | 2630 (29.81) | 621 (39.33) | 2009 (27.74) | | 1834 (27.98) | 458 (35.59) | 1376 (26.12) | |
| Hospital | 2353 (26.67) | 502 (31.79) | 1851 (25.56) | | 1843 (28.12) | 401 (31.16) | 1442 (27.38) | |
| General hospital | 916 (10.38) | 191 (12.10) | 725 (10.01) | | 573 (8.74) | 145 (11.27) | 428 (8.13) | |
| Tertiary hospital | 44 (0.50) | 18 (1.14) | 26 (0.36) | | 53 (0.81) | 22 (1.71) | 31 (0.59) | |
| Oriental medicine hospital | 1057 (11.98) | 135 (8.55) | 922 (12.73) | | 1077 (16.43) | 170 (13.21) | 907 (17.22) | |
| **30-day readmission** | **3252 (4.15)** | **766 (3.20)** | **2486 (4.56)** | **< .0001** | **3036 (3.79)** | **853 (3.12)** | **2183 (4.13)** | **< .0001** |
| The same hospital | 1366 (42.00) | 159 (20.76) | 1207 (48.55) | < .0001 | 1197 (39.43) | 166 (19.46) | 1031 (47.23) | < .0001 |
| Other hospitals | 1886 (58.00) | 607 (79.24) | 1279 (51.45) | | 1839 (60.57) | 687 (80.54) | 1152 (52.77) | |
| **Type of healthcare center in readmission** | **n = 3252** | **n = 766** | **n = 2486** | **< .0001** | **n = 3036** | **n = 853** | **n = 2183** | **< .0001** |
| Convalescent hospital | 195 (6.00) | 8 (1.04) | 187 (7.52) | | 160 (5.27) | 10 (1.17) | 150 (6.87) | |
| Public health center | 4 (0.12) | 0 (0.00) | 4 (0.16) | | 0 (0.00) | 0 (0.00) | 0 (0.00) | |
| Clinic | 351 (10.79) | 69 (9.01) | 282 (11.34) | | 330 (10.87) | 75 (8.79) | 255 (11.68) | |
| Hospital | 1821 (56.00) | 546 (71.28) | 1275 (51.29) | | 1710 (56.32) | 603 (70.69) | 1107 (50.71) | |
| General hospital | 508 (15.62) | 95 (12.40) | 413 (16.61) | | 454 (14.95) | 104 (12.19) | 350 (16.03) | |
| Tertiary hospital | 121 (3.72) | 27 (3.52) | 94 (3.78) | | 111 (3.66) | 39 (4.57) | 72 (3.30) | |
| Oriental medicine hospital | 252 (7.75) | 21 (2.74) | 231 (9.29) | | 271 (8.93) | 22 (2.58) | 249 (11.41) | |

*LOH*, length of hospital stay.

## Discharge disposition

In the pre-COVID-19 cohort, 11.25% of patients were transferred to other hospitals after discharge, while this was the case for 8.18% of patients in the COVID-19 cohort (p<0.01). A significantly higher proportion of patients with longer hospital stays were transferred to another hospital than that of patients with hospital days of 7 days or less (p<0.01 in both cohorts). The most common type of transfer location was a clinic, followed by a primary hospital, convalescent hospital, oriental medicine hospital, general hospital, tertiary hospital, and public health center. The order was the same in the pre-COVID-19 cohort, irrespective of LOH. However, in the COVID-19 cohort, the order of clinics and primary hospitals was switched (Table 4).

## Thirty-day readmission and reoperation

The 30-day readmission rate was 4.15% in the pre-COVID-19 cohort and 3.79% in the COVID-19 cohort. In both cohorts, the rates were higher in patients with longer LOH (p<0.01 in both the pre-COVID-19 and COVID-19 cohorts). Regarding readmission, in both cohorts, approximately 40% of patients were re-admitted to the same hospital, while around 60% of patients were re-admitted at other hospitals. More patients with LOH ≥ 8 days were

**Table 5. Readmissions, reoperations, and transfers.**

| | Number of patients | Number of events | Crude odds ratio (95% CI) | p-value | Adjust odds ratio* (95% CI) | p-value |
|---|---|---|---|---|---|---|
| **30-day readmission** | | | | | | |
| Pre-COVID-19 | 78402 | 3252 (4.15) | 1 (ref) | | 1 (ref) | |
| COVID-19 | 80165 | 3036 (3.79) | 0.910 (0.865,0.957) | 0.0002 | 0.924 (0.879,0.972) | 0.0023 |
| **30-day reoperation** | | | | | | |
| Pre-COVID-19 | 78402 | 704 (0.90) | 1 (ref) | | 1 (ref) | |
| COVID-19 | 80165 | 779 (0.97) | 1.083 (0.978,1.200) | 0.1269 | 1.049 (0.946,1.162) | 0.3639 |
| **Transfer to other hospital** | | | | | | |
| Pre-COVID-19 | 78402 | 8822 (11.25) | 1 (ref) | | 1 (ref) | |
| COVID-19 | 80165 | 6554 (8.18) | 0.702 (0.679,0.726) | < .0001 | 0.715 (0.691,0.740) | < .0001 |

* Adjusted for CCI, sex, age, income level, length of hospital stay, Parkinson's disease, osteoporosis, type of insurance, and type of hospital.

readmitted to the same hospital (48.55% in the pre-COVID-19 cohort and 47.23% in the COVID-19 cohorts) than patients with LOH $\leq$ 7 days (20.76% in the pre-COVID-19 cohort [p<0.01] and 19.46% in the COVID-19 cohort [p<0.01]). The most common type of health-care institution was primary hospitals in both the pre-COVID-19 and COVID-19 cohorts (Table 4). The reoperation rate was 0.90% in the pre-COVID-19 cohort and 0.97% in the COVID-19 cohort (Table 5). The time to reoperation was 17.28 ± 7.82 days in the pre-COVID-19 cohort and 17.11 ± 7.88 days in the COVID-19 cohort.

Compared to the pre-COVID-19 period, there was a significant decrease in readmissions within 30 days and transfers to other hospitals (adjusted OR (aOR): 0.92, 95% confidence interval (CI): 0.88–0.97, p<0.01; aOR: 0.72, 95% CI: 0.69–0.74, p<0.01). However, no significant increase or decrease was observed in reoperations after surgery during this time (aOR: 1.05, 95% CI: 0.95–1.16, p = 0.36) (Table 5).

## Discussion

The primary aim of our study was to investigate trends in lumbar spinal surgery during the first year of the COVID-19 pandemic in the ROK, especially in relation to specific epidemic waves. Notably, while the total number of surgeries remained stable during the pandemic, there were noticeable decrease in the monthly operation counts influenced by the early epidemic waves. These initial decreases were later balanced by an increase in surgeries in the following months. The joinpoint regression effectively captured the decline influenced by the early epidemic waves, which was then followed by a compensatory rise. Interestingly, this pattern of a decrease followed by a compensatory increase was absent during the third epidemic wave in November 2020.

After the WHO declared COVID-19 a pandemic in March 2020, most elective procedures were postponed worldwide in order to care for the unprecedented influx of critically ill patients [12,15]. Ridwan et al. reported that, in Europe, only 15% of respondents in the European Association of Neurosurgical Societies (EANS) replied that elective operations continued during April and May 2020 [31]. Blondel et al. reported that, in France, operating rooms for degenerative spine surgery operated at 20 to 50% of normal capacity, with the SFCR (The French Society of Spinal Surgery) guidelines during the early 2020 [12]. In the US, after the suspension of spinal surgery from March to June 2020, 47% of 133 patients were re-scheduled by June 2020, 16% were re-scheduled by January 2021, and 37% were ultimately not re-scheduled. Norris et al. also showed that overall, one third of patients whose surgery was suspended at the beginning of the mandatory suspension of elective surgery ultimately did not receive

surgical treatment [14]. Patients with neurological dysfunction or severe symptoms were re-scheduled once the mandatory suspension of elective operations was lifted [14,32], but about 40% of patients without those problems did not undergo elective surgery.

In contrast, the situation in ROK during the first year of the pandemic showed distinct differences. First, the country experienced a relatively low number of COVID-19 cases compared to others, despite three specific epidemic waves. Second, unlike some countries, the Korean government did not implement regulatory restrictions on elective surgeries during the pandemic. Tertiary hospitals were mostly responsible for treating COVID-19 patients, allowing private hospitals to maintain their surgical volume in terms of medical resources. This redistribution may explain the decreased proportion of elective lumbar surgeries performed in tertiary hospitals during the pandemic (Table 1). Nevertheless, the unprecedented pandemic itself discouraged patients from undergoing elective surgery, particularly during the periods of specific epidemic waves.

## Number of operations, monthly distribution, and epidemic waves

In ROK, three epidemic waves occurred in the first year of COVID-19 pandemic, and the peak of each wave reached on February 29th, August 27th, and December 25th, 2020. The numbers of COVID-19 patients and elective spine surgical procedures by month in 2020 showed an overall inverse correlation with the number of COVID-19 patients and the number of operations, reflecting natural response of patients and physicians to the epidemic waves. When the first wave broke out in February, the number of spinal operations decreased significantly ($p<0.01$). However, after April, as the number of COVID-19 patients stabilized, the number of spine operations increased again, reaching a level similar to that before COVID-19 in May and a higher level in June. In August, the second wave occurred, which was followed by a decrease in spine operations in September ($p = 0.21$). However, when the number of COVID-19 patients decreased after that wave, the number of spine operations then increased again in October, this time more quickly than after the first epidemic wave. Lastly, the third wave, the most serious ever, occurred in November, but there was no significant decrease in spinal surgery, unlike in the previous two waves (Fig 1).

These results demonstrate a gradual weakening of the inverse correlation between the number of COVID-19 patients and the number of elective spine operations as the pandemic progressed through multiple waves. The results of the present study may imply that the know-how obtained from past waves could have reduced patients' concerns. Moreover, patients who initially postponed surgery due to concerns about COVID-19 infection may no longer have been able to tolerate pain or disability as the pandemic persisted [14,32,33]. This could explain the observed increase in elective spine surgery during the surge of COVID-19 cases in November and December.

The current study showed that even during the COVID-19 pandemic, total number of elective surgeries for degenerative lumbar spinal disease was maintained, despite the impact of epidemic waves. This result implies that with appropriate allocation of medical resources and medical policies, the core criteria for surgical indications and elective surgery decision-making should be unchanged. Some patients may have exhibited spontaneous improvement during the waiting period, but some may have chosen not to undergo surgery despite their disability not improving. Even those who experienced spontaneous improvement might have benefitted from surgery (e.g., less pain, potentially better neurologic recovery, etc.). Consequently, it is possible that certain patients continued to suffer from pain and disability and were unable to return to their normal lives [14,34–36]. With applying surgical criteria strictly to decide the necessity of elective lumbar surgeries, essential elective surgeries could be performed. This

lesson may be useful in planning policies about spine surgery and transitioning to the "new normal" era.

## Changes during hospital stays

**Types of surgery, LOH, and hospital costs.** Overall, the distribution of surgical procedures did not substantially change. The distribution of surgical procedures for HIVD, however, saw an increase in discectomy and endoscopic discectomy, and a decrease in fusion surgery, which may suggest a pandemic-driven preference for less invasive procedures over more complex surgeries, likely due to concerns regarding extended hospital stays and postoperative care [14,36–39]. Conversely, for LSS and spondylolisthesis, the proportion of fusion surgeries increased, which is inconsistent with the presumed preference toward less invasive surgery. A contributing factor may have been that the reduction in physical activity during the pandemic potentially caused exacerbating symptoms of LSS and spondylolisthesis and prompting patients to proceed with more definitive surgical interventions. The overall increase in surgeries for LSS and spondylolisthesis could also be reflective of this exacerbation. Nonetheless, since our dataset lacks detailed clinical information, the exact cause of the change remains unexplainable.

This study showed a reduced LOS during the COVID-19 pandemic. This aligns with findings from multiple studies across different types of surgery which reported a similar reduction in LOH during the pandemic, including adult spinal deformity surgery [40], elective colorectal surgery [41,42], brain tumor surgery [43], bariatric surgery [44], and partial nephrectomy [45] without compromising surgical outcomes. Several factors may have contributed to this decrease. The pandemic increased the demand for hospital beds and shortening the hospital stay was a strategy to maintain the volume of surgery [44]. Patients may have also wanted to limit their hospital stays to reduce the risk of exposure to COVID-19 [40,45]. Moreover, patients were also more willing to follow hospitals' recommendations than before the COVID-19 pandemic [40,46]. Additionally, the increase in medical costs may have influenced the LOH. Those could be reasons for shortened LOH in countries with national health insurance systems similar to that of ROK. However, hospital costs went up despite the shortened LOH. Between 2019 and 2020, the costs of each surgical procedure increased by 10%, but hospital costs increased by approximately 16% (Table 3). This intriguing finding, however, is limited by the unavailability of detailed hospital cost data for this study. Several hypotheses could explain this, such as the expenses related to additional COVID-19 diagnostic tests, a preference for single occupancy rooms, shortages in medical resources, increased costs of goods and services, and inflation. Further research is required to pinpoint the precise reasons behind the increase in hospital costs.

## Changes after discharge

**Discharge disposition, readmission rate, and reoperation rate.** More patients were discharged to their homes in the COVID-19 cohort. During the COVID-19 pandemic, patients and hospitals were reluctant to transfer patients to other hospitals due to concerns about cross-nosocomial infection [14,40,47,48]. This may have been a reason for the decreased proportion of hospital discharges. In both the pre-COVID-19 and COVID-19 clinics, patients were commonly transferred to primary hospitals and clinics. This finding implies that the basic roles of each hospital were maintained even during the COVID-19 pandemic. It was interesting that the shortened LOH and increased proportion of home discharges did not increase the 30-day readmission rate or the reoperation rate. The surgical technique and care protocol were similar between the pre-COVID-19 and COVID-19 cohorts, and the 1-day

shorter hospital stay may not have changed the 30-day readmission rate. However, limited access to hospitals for less essential or emergent medical problems may have raised the threshold for readmission criteria. Wang et al. demonstrated that the major complication rate did not change during the COVID-19 pandemic despite shorter hospital stays [40]. Mohammed et al. also showed that complications rate did not increase during the COVID-19 pandemic [15]. However, this should not be interpreted as meaning that patients had the same conditions at discharge and during the postoperative 30-day period [46]. Pinloche et al. showed that patients had shorter stays, but were discharged with more intense pain during the COVID-19 pandemic than before [46]. Despite the similar 30-day readmission and reoperation rate, limited rehabilitation due to lockdowns may have influenced clinical outcomes [46]. Moreover, the thresholds of patients and doctors might have changed, and patients and doctors might have been reluctant to readmit patients unless their conditions were severe enough to require reoperation during the COVID-19 pandemic. Further studies with clinical data and patient satisfaction rates are required. Nonetheless, this study showed that elective spinal surgery could be continued without an increase in the readmission or reoperation rates. Maintaining the basic role of healthcare providers, while considering capacity for critical COVID-19 patients, would be helpful for patients requiring elective spinal surgery [15,40,46].

### Health care system and infrastructure

The current results were obtained from data of ROK, where all people are beneficiary of national health insurance or medical aid. The access to any hospital is not limited to place, disease, and type of insurance. In the ROK, a significant majority of degenerative lumbar spine surgeries are performed in non-tertiary hospitals (approximately 90%, Table 1). Therefore, securing hospital beds for COVID-19—predominantly at tertiary hospitals—did not markedly reduce the capacity for treating patients with degenerative lumbar spinal disease. In this regard, a catch-up increase of surgeries was possible with Korean health insurance system without disrupting urgent care for the COVID-19 patients. However, the proportion of patients in each type of hospital may be different with other disease such as cancer or other complex disease. Therefore, the current idea of allowing or modifying the quota of elective surgeries may be customized based on similar analysis with each disease. In addition, the proportion of degenerative lumbar spine surgeries in large hospitals, where most of COVID-19 patients were cared, may be various according to countries and health care system. The current study showed the idea of using secondary data to make plan for contingency and it should not be a generalized plan to every disease, country and health care system.

### Limitations

Although this study demonstrated the impact of the COVID-19 pandemic on lumbar degenerative disease within ROK's national health insurance system, it had several limitations. First, the secondary data lacked information on clinical characteristics, and the exact causes of change in surgical procedures, LOH, and increased costs could not be established. The unresolved issue warrants further analysis of clinical data. In particular, changes in hospital and surgical protocols during the pandemic period could act as potential unmeasured confounders, although we were not able to identify specific changes. Second, the lessons from this study may not be generalizable because the strategies for managing the COVID-19 pandemic may have differed among countries and health care system. Nonetheless, although the specific strategies varied, the basic principle of controlling spread of infection was the same. In this regard, this study suggests that mandatory suspension of elective spinal surgery in all hospitals may not be necessary considering the specific role of each hospital according to the situation in

each country. Third, during the COVID-19 pandemic, the entire world was attentive and worked together to contain the pandemic. However, such unanimous action may not be guaranteed in the future, because the knowledge gained from the COVID-19 pandemic varies among countries. Therefore, the experiences from this study may not be applicable in every situation. Despite these limitations, this study showed that ROK's efforts to curb the COVID-19 pandemic were successful in maintaining the quality and quantity of spinal surgery. Although we have faced challenges during the COVID-19 pandemic, it is important to share the valuable lessons during these difficult times to avoid experiencing the same chaos in the future.

## Conclusions

Throughout the early stages of the COVID-19 pandemic, our study observed that while elective lumbar spinal surgeries were initially disrupted by pandemic and epidemic waves, they subsequently experienced a compensatory increase, indicating that surgical care adapted to the pandemic's challenges. This adaptability, alongside the stable total number of operations, highlights the potential for healthcare systems to continue essential surgical services during public health crises with strategic resource allocation and patient triage. As we consider the indirect effects of the pandemic on hospital length of stay, discharge dispositions, costs, and readmission rates, these insights underscore the importance of developing robust, flexible healthcare policies for clinicians, researchers, and healthcare policy makers. Such policies should ensure that elective spinal surgeries, especially those not requiring urgent care but crucial for patient quality of life, are not unnecessarily halted. Although circumstances vary from country to country, making it difficult to generalize these findings universally, the valuable lessons learned from the challenges of the COVID-19 pandemic should be used as a guide for future strategies in managing similar health crises.

## Acknowledgments

The authors appreciate the statistical advice provided by the Medical Research Collaborating Center at Seoul National University Hospital.

## Author Contributions

**Conceptualization:** Chi Heon Kim.

**Data curation:** Jinhee Kim, Mi-Sook Kim, Young Rak Kim, Sum Kim.

**Formal analysis:** Jinhee Kim, Mi-Sook Kim, Jun-Hoe Kim, Young Rak Kim, Sum Kim.

**Funding acquisition:** Chi Heon Kim.

**Investigation:** Mi-Sook Kim, Chi Heon Kim.

**Methodology:** Jinhee Kim, Mi-Sook Kim, Chun Kee Chung, Sung Bae Park, John M. Rhee, Chi Heon Kim.

**Supervision:** Chun Kee Chung, Chang-Hyun Lee, Kyoung-Tae Kim, John M. Rhee, Young San Ko, Chi Heon Kim.

**Validation:** Jinhee Kim, Young San Ko.

**Visualization:** Woon Tak Yuh.

**Writing – original draft:** Woon Tak Yuh, Chi Heon Kim.

**Writing – review & editing:** Woon Tak Yuh, Mi-Sook Kim, Jun-Hoe Kim, Young Rak Kim, Sum Kim, Chun Kee Chung, Chang-Hyun Lee, Sung Bae Park, Kyoung-Tae Kim, John M. Rhee, Young San Ko, Chi Heon Kim.

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
