## [Decision Letter · Decision Letter 0]

29 Feb 2024

PONE-D-23-41698Trends in degenerative lumbar spinal surgery during the early COVID-19 pandemic in Republic of Korea: a national study utilizing the National Health Insurance DatabasePLOS ONE

Dear Dr. Kim,

Thank you for submitting your manuscript to PLOS ONE. After careful consideration, we feel that it has merit but does not fully meet PLOS ONE’s publication criteria as it currently stands. Therefore, we invite you to submit a revised version of the manuscript that addresses the points raised during the review process.

We look forward to receiving your revised manuscript.

Kind regards,

Kentaro Yamada, M.D., Ph.D.

Academic Editor

PLOS ONE

Journal Requirements:

This study was supported by Seoul National University Hospital Research Fund (Grant No. 3020230120), Doosan Yonkang Foundation (Grant No. 800-20210527), and Armed Forces Capital Hospital (Grant No. 2023MDD0075), received by Chi Heon Kim. The funders had no role in study design, data collection and analysis, decision to publish, or preparation of the manuscript.

Reviewers' comments:

Reviewer's Responses to Questions

**Comments to the Author**

1. Is the manuscript technically sound, and do the data support the conclusions?

Reviewer #1: Yes

Reviewer #2: Partly

2. Has the statistical analysis been performed appropriately and rigorously? 

Reviewer #1: Yes

Reviewer #2: No

3. Have the authors made all data underlying the findings in their manuscript fully available?

Reviewer #1: No

Reviewer #2: Yes

4. Is the manuscript presented in an intelligible fashion and written in standard English?

Reviewer #1: Yes

Reviewer #2: Yes

5. Review Comments to the Author

Reviewer #1: Thank you for submitting this interesting and informative manuscript to PLoS One. I was pleased to receive it as a reviewer.

While your manuscript provides valuable insights into an important healthcare topic, there are a few areas that could be refined to further augment the quality and impact of the work. Here are some respectful suggestions that could potentially improve the paper if you choose to implement them:

Abstract

- The abstract clearly summarizes key aspects, but the conclusion may be strengthened by elaborating on the practice-relevant implications regarding elective surgery policies during pandemics.

Introduction

- The introduction effectively covers the background of the study. As the discussion notes study limitations regarding generalizability, the introduction could benefit from briefly mentioning upfront that the healthcare infrastructure and pandemic responses likely differ across countries.

- Consider further elaborating on the rationale for focusing specifically on degenerative lumbar spinal diseases and surgeries. For example, you could explain if patients with these conditions could be more vulnerable during pandemics versus other spine issues.

Methods

- The methods are detailed overall. As a suggestion, it could be useful to state if there were any changes to hospital or surgical protocols between the cohorts that could have impacted the results through unmeasured confounders.

- Consider providing more details on the propensity score matching process. Which specific variables were included and why? How was the greedy matching algorithm optimized?

- Consider explaining the rationale for replacing surgery numbers for holiday weeks with average values. Elaborate on assumptions and implications.

Results

- When presenting the changes in proportions of surgery types for the diagnoses, consider speculating on potential reasons driving the shifts. This would give readers more context when interpreting the results.

Discussion

- The discussion provides meaningful interpretation of results and balances the strengths against limitations. To augment the actionability of the conclusions, you could elaborate further on the presumed practice-relevant implications regarding elective surgery decisions during pandemics.

- Compare and contrast findings with other COVID-19-era studies on spine surgery trends from different countries to put results into perspective. How and why are they similar or different?

- Discuss the degree to which the conclusions can be reasonably extrapolated to healthcare systems with differing infrastructure and pandemic responses compared to the Republic of Korea population.

Conclusions

- The conclusions concisely summarize the main outcomes. To amplify the impact, you may consider incorporating more specific guidance regarding elective spine surgery policies during future public health emergencies based on your findings.

Overall, these suggestions aim to enhance the manuscript's overall quality for clinicians, researchers, and healthcare policy makers. I believe that implementing the above suggestions would make your important work more useful for informing practice under crisis situations.

Reviewer #2: Thank you for the opportunity to review the paper by Kim and colleagues regarding a study on the status of spinal surgery during the COVID-19 pandemic in South Korea using real-world data. This is a well-written manuscript and offers valuable societal insights. I read this manuscript with great interest. While I respect the authors' significant effort in extracting detailed information from nationwide population-based data, there are several methodological problems with this study.

Major Comments:

1. The utilization of propensity score matching as a method in this study appears inappropriate, given the assumptions associated with this technique. If the intention is to compare pre- and post-population, I would recommend employing regression adjustment incorporating variables pertinent to the outcome of interest. As alternative methods, standardization methods such as Standardized Mortality Ratios (SMR) could be considered.

2. Overall, the paper is redundant in its description. For example, the "patient inclusion" and "variables" sections have room for improvement.

3. (Page 7, line 157-) Although many populations have been excluded, these exclusions may be inappropriate when examining trends in the number of surgeries.

4. (Page 15, line 253-) In applying joinpoint regression, the authors appear to have corrected the data, but I cannot determine if this is appropriate. I'm afraid this data handling may affect the results.

Minor comments

1. Many studies have pointed to a decrease in postoperative hospital stay for various surgeries. Consider citing these.

2. (Table 1) The column of CCI is broken; please correct it.

Finally, I would like to express my respect for the great effort of the authors in this paper. I hope this paper will be properly revised and published.

6. PLOS authors have the option to publish the peer review history of their article (what does this mean?). If published, this will include your full peer review and any attached files.

Reviewer #1: **Yes: **Savvas Lampridis

Reviewer #2: No

---

## [Author Response · Author response to Decision Letter 0]

15 Apr 2024

Reviewer #1

Comment - Thank you for submitting this interesting and informative manuscript to PLOS ONE. I was pleased to receive it as a reviewer.

Authors’ response:

Thank you for your time and effort to review our manuscript. We truly appreciate your valuable comments. 

Comment - While your manuscript provides valuable insights into an important healthcare topic, there are a few areas that could be refined to further augment the quality and impact of the work. Here are some respectful suggestions that could potentially improve the paper if you choose to implement them:

Authors’ response: 

We are thankful for your insightful comments, which have significantly contributed to the enhancement of our manuscript, particularly in areas concerning healthcare system and infrastructure, policies across different countries, clinical implications, and future healthcare crisis strategy. We wholeheartedly agree with your perspective. It was indeed a great pleasure to revise our manuscript in line with your valuable suggestions, and we hope that our revisions will fulfill your expectations and meet the standards for publication.

Abstract

Comment #1 - The abstract clearly summarizes key aspects, but the conclusion may be strengthened by elaborating on the practice-relevant implications regarding elective surgery policies during pandemics.

Authors’ response:

Thank you for your constructive feedback on our abstract. We appreciate your suggestion to elaborate on the practice-relevant implications of elective surgery policies during pandemics. During the pandemic, COVID-19 patients were primarily treated in tertiary hospitals, allowing lumbar surgeries to proceed with minimal disruption in other types of healthcare facilities. This pattern, while not initially a strategic approach, demonstrated an effective allocation of medical resources under the national health insurance system. These findings, emerging from the challenges posed by the pandemic, offer valuable insights for future planning. Employing such a triage approach could serve as a deliberate strategy in managing elective surgeries during future health crises. In our revised conclusion, we have highlighted the observed effect, which acted like a natural triage system. 

Change to Text:

[Abstract, Page 3, Line 63-69] 

Despite the impact of epidemic waves on monthly surgery numbers, a subsequent compensatory increase was observed, indicating that surgical care has adapted to the challenges of the pandemic. This adaptability, along with the stable total number of operations, highlights the potential for healthcare systems to continue elective spine surgery during public health crises with strategic resource allocation and patient triage. Policies should ensure that surgeries for degenerative spinal diseases, particularly those not requiring urgent care but crucial for patient quality of life, are not unnecessarily halted.

Introduction

Comment #2 - The introduction effectively covers the background of the study. As the discussion notes study limitations regarding generalizability, the introduction could benefit from briefly mentioning upfront that the healthcare infrastructure and pandemic responses likely differ across countries.

Authors’ response:

Thank you for your comment regarding the need to clarify the context of healthcare infrastructure and pandemic responses across countries. We acknowledge that our findings are specific to the ROK's healthcare system, which is characterized by universal coverage through national health insurance and medical aid. We have included a brief mention of the ROK's healthcare infrastructure, which, while not universally applicable, serves as a conceptual framework for planning in diverse healthcare systems. We also have included the distinct pandemic response of ROK, different from other countries. 

Change to Text:

[Introduction, Page 4, Line 81-89] 

While the main guidelines and strategies in every country were similar, the healthcare infrastructure and pandemic responses, including spinal surgery varied among countries. [12, 13] For example, in ROK, all people are beneficiary of national health insurance or medical aid, and the access to any hospital is not limited to place. During the COVID-19 pandemic, the government of ROK did not restrict elective surgical procedures officially. On the other hand, in the USA, the COVID-19 pandemic led to a nationwide suspension of elective surgery, including spinal procedures, from March 2020 to May 2020.[14] the COVID-19 pandemic led to a nationwide suspension of elective surgery, including spinal procedures, from March 2020 to May 2020.[14]

Although the government of ROK did not officially restrict elective spinal surgeries during the pandemic, the conditions for such surgeries changed significantly.

Comment #3 - Consider further elaborating on the rationale for focusing specifically on degenerative lumbar spinal diseases and surgeries. For example, you could explain if patients with these conditions could be more vulnerable during pandemics versus other spine issues.

Authors’ response:

We are grateful for the opportunity to clearly explain the focus of our study on degenerative lumbar spinal diseases and surgeries. As briefly mentioned in the introduction part of the original manuscript, degenerative lumbar surgeries are performed not for treating an immediate threat to life, but for improving patients’ quality of life, by alleviating pain and functional impairment. However, during a pandemic, when medical priorities shift towards acute critical care for COVID-19 patients, these lumbar elective surgeries are prone to delays or cancellations, consequently rendering them significantly vulnerable. Even compared to other spine issues, such as cervical or thoracic spinal disease, or spinal trauma, the medical priority of degenerative lumbar disease is low. While postponements may seem appropriate considering the short-term urgency of a pandemic, they can cause increased suffering for patients with degenerative spine issues as the pandemic becomes protracted, even if their conditions are not directly life-threatening. We have further elaborated on these aspects in the introduction part as suggested by the reviewer.

Change to Text:

[Introduction, Page 5, Line 99-109]

Surgical interventions for degenerative lumbar spinal disease are typically elective and focused on enhancing quality of life rather than addressing emergencies that threaten life immediately. As a result, these conditions are often assigned a lower priority compared to other spinal diseases such as cervical or thoracic lesions with neurological deficits or spinal traumas requiring urgent surgical interventions. In the initial stages of the COVID-19 pandemic, elective surgeries for degenerative lumbar diseases were frequently deferred or canceled as medical priorities shifted to pandemic management. This practice, although intended to conserve medical resources for acute pandemic care, was found to potentially worsen the long-term quality of life of these patients, increasing their vulnerability as the pandemic protracted. [12, 14-19] Based on this experience, this study was designed to observe the impact of the COVID-19 pandemic on surgeries for patients with degenerative lumbar spinal disease who are vulnerable during pandemics.

Methods

Comment #4 - The methods are detailed overall. As a suggestion, it could be useful to state if there were any changes to hospital or surgical protocols between the cohorts that could have impacted the results through unmeasured confounders.

Authors’ response:

Thank you for your comment regarding the potential impact of changes in hospital or surgical protocols on our results. We analyzed the NHID customized data, representing a claim dataset of entire national population. Although it lacks detailed clinical data, our objective was to examine the trends of elective lumbar surgery influenced by numerous COVID-19 pandemic-related factors, including any modifications to hospital or surgical protocols. We specifically looked for changes in hospital or surgical protocols unrelated to the COVID-19 pandemic that could serve as significant confounders. However, we couldn’t find any protocol change between the cohorts during the study period that would qualify as unmeasured confounders separate from those associated with the COVID-19 pandemic. We have acknowledged this in the limitations, noting the potential for confounders related to changes in hospital and surgical protocols, although we were not able to identify specific changes. Again, thank you for raising an important question.

Change to Text:

[Limitation, Page 25, Line 483-487]

First, the secondary data lacked information on clinical characteristics, and the exact causes of change in surgical procedures, LOH, and increased costs could not be established. The unresolved issue warrants further analysis of clinical data. In particular, changes in hospital and surgical protocols during the pandemic period could act as potential unmeasured confounders, although we were not able to identify specific changes.

Comment #5 - Consider providing more details on the propensity score matching process. Which specific variables were included and why? How was the greedy matching algorithm optimized?

Authors’ response:

Thank you for your question concerning the details of the propensity score matching process. Initially, our model employed propensity score matching to compare the pre- and post-COVID-19 cohorts. However, based on the reviewers' feedback, highlighting concerns about the appropriateness of the propensity score matching, and on suggestion from another reviewer, we decided to change our comparison approach to linear regression in the whole population with multivariable adjustment. This revised method could be more suitable for comparison between the pre-COVID-19 and COVID-19 periods while enabling us to effectively control for potential confounders. We believe that this modification better aligns with the study's objectives.

Change to Text:

[Materials and methods, Page 13, Line 246-250]

A logistic regression model was used to compare the rates of reoperation and readmission within 30 days after surgery between pre-COVID-19 and COVID-19 periods. For multivariate analysis, adjustments were made for age, sex, CCI, income level, length of hospital stay, Parkinson’s disease, osteoporosis, type of insurance, and type of hospital. Crude and adjusted odds ratios (ORs), along with their 95% confidence intervals, were estimated.

Comment #6 - Consider explaining the rationale for replacing surgery numbers for holiday weeks with average values. Elaborate on assumptions and implications.

Authors’ response:

Thank you for your comment. Our rationale for initially replacing surgery numbers during holiday weeks with average values was to minimize the impact of reduced medical service utilization during the Lunar New Year and Thanksgiving. In Korea, these holidays are traditionally accompanied by at least four consecutive days off, and it's common for people to avoid scheduling elective surgeries during these times. However, acknowledging your concern about how this approach might affect the results, we revised our methodology. We no longer replace surgery numbers for holiday weeks with average values; instead, we adjust the weekly number of surgeries by working day. This adjustment would provide a more accurate reflection of surgical activity during these periods without altering the data.

Change to Text:

[Materials and methods, Page 12, Line 233-240]

Joinpoint regression analysis was applied to examine changes in time trends of surgical cases over the first year of the COVID-19 pandemic period. This statistical method identifies the time points, called "joinpoints", where linear time trend significantly change and calculated the percentage change between trend-change points, along with the average percentage change in the entire period. The number of spinal surgeries was counted on a weekly basis from January to December 2020, and a 3-week moving average was calculated. To account for the decrease in surgical cases due to weekends or public holidays, such as Lunar New Year and Thanksgiving, the weekly number of surgical cases was adjusted based on the number of working days in those weeks.

Results

Comment #7 - When presenting the changes in proportions of surgery types for the diagnoses, consider speculating on potential reasons driving the shifts. This would give readers more context when interpreting the results.

Authors’ response:

Thank you for your valuable comment. In the discussion section of our manuscript, we have briefly mentioned the potential reasons for the shifts in surgery types. Overall, the distribution of surgical procedures did not substantially change. The distribution of surgical procedures for HIVD, however, saw an increase in discectomy and endoscopic discectomy, and a decrease in fusion surgery, which may suggest a pandemic-driven preference for less invasive procedures over more complex surgeries, likely due to concerns regarding extended hospital stays and postoperative care. 

Conversely, for LSS and spondylolisthesis, the proportion of fusion surgeries increased, which is inconsistent with the presumed preference toward less invasive surgery. A contributing factor may have been that the reduction in physical activity during the pandemic potentially caused exacerbating symptoms of LSS and spondylolisthesis and prompting patients to proceed with more definitive surgical interventions. The overall increase in surgeries for LSS and spondylolisthesis could also be reflective of this exacerbation. Nonetheless, our dataset lacks detailed clinical information to explain the exact causes of these shifts. Hence, while we can offer plausible hypotheses, the precise reasons for the changes in surgical trends remain elusive. We have included this context in the discussion part.

Change to Text:

[Discussion, Page 22, Line 409-419]

Overall, the distribution of surgical procedures did not substantially change. The distribution of surgical procedures for HIVD, however, saw an increase in discectomy and endoscopic discectomy, and a decrease in fusion surgery, which may suggest a pandemic-driven preference for less invasive procedures over more complex surgeries, likely due to concerns regarding extended hospital stays and postoperative care.[14, 36-39] Conversely, for LSS and spondylolisthesis, the proportion of fusion surgeries increased, which is inconsistent with the presumed preference toward less invasive surgery. A contributing factor may have been that the reduction in physical activity during the pandemic potentially caused exacerbating symptoms of LSS and spondylolisthesis and prompting patients to proceed with more definitive surgical interventions. The overall increase in surgeries for LSS and spondylolisthesis could also be reflective of this exacerbation. Nonetheless, since our dataset lacks detailed clinical information, the exact cause of the change remains unexplainable.

Discussion

Comment #8 - The discussion provides meaningful interpretation of results and balances the strengths against limitations. To augment the actionability of the conclusions, you could elaborate further on the presumed practice-relevant implications regarding elective surgery decisions during pandemics.

Authors’ response: 

Thank you for your insightful suggestion. We recognize that the core criteria for surgical indications and elective surgery decision-making remain unchanged, while the allocation of medical resources and patients' attitudes toward elective surgery become critical in a pandemic setting. We advocate for the application of strict surgical criteria to decide the necessity of elective lumbar surgeries, thereby ensuring that only essential surgeries are performed. This approach should take into account the available medical resources to prioritize effectively among all patients, including those with COVID-19. 

Change to Text:

[Discussion, Page 21, Line 394-405]

The current study showed that even during the COVID-19 pandemic, total number of elective surgeries for degenerative lumbar spinal disease was maintained, d

---

## [Decision Letter · Decision Letter 1]

8 May 2024

PONE-D-23-41698R1Trends in degenerative lumbar spinal surgery during the early COVID-19 pandemic in Republic of Korea: a national study utilizing the National Health Insurance DatabasePLOS ONE

Dear Dr. Kim,

Thank you for submitting your manuscript to PLOS ONE. After careful consideration, we feel that it has merit but does not fully meet PLOS ONE’s publication criteria as it currently stands. Therefore, we invite you to submit a revised version of the manuscript that addresses the points raised during the review process.

We look forward to receiving your revised manuscript.

Kind regards,

Kentaro Yamada, M.D., Ph.D.

Academic Editor

PLOS ONE

Journal Requirements:

**Additional Editor Comments:**

Thank you for your lots of efforts to improve article. Please confirm comments from Reviewer #2, then reply for the 2 questions.

Reviewers' comments:

Reviewer's Responses to Questions

**Comments to the Author**

1. If the authors have adequately addressed your comments raised in a previous round of review and you feel that this manuscript is now acceptable for publication, you may indicate that here to bypass the “Comments to the Author” section, enter your conflict of interest statement in the “Confidential to Editor” section, and submit your "Accept" recommendation.

Reviewer #1: All comments have been addressed

Reviewer #2: All comments have been addressed

2. Is the manuscript technically sound, and do the data support the conclusions?

Reviewer #1: Yes

Reviewer #2: Yes

3. Has the statistical analysis been performed appropriately and rigorously? 

Reviewer #1: Yes

Reviewer #2: Yes

4. Have the authors made all data underlying the findings in their manuscript fully available?

Reviewer #1: No

Reviewer #2: Yes

5. Is the manuscript presented in an intelligible fashion and written in standard English?

Reviewer #1: Yes

Reviewer #2: Yes

6. Review Comments to the Author

Reviewer #1: Thank you for the attention and consideration you have shown to my suggested revisions for your manuscript. It is evident that a significant amount of effort and thought has been directed towards the refining of your work, integrating the feedback provided during the peer review process. The resulting modifications demonstrate a thorough approach and significantly improve the rigor and overall quality of your manuscript. I look forward to witnessing the impact your research will undoubtedly have on the academic community.

Reviewer #2: I would like to thank the authors for their considerable effort regarding the revision. My previous comments were appropriately addressed.

Therefore, my decision is provisional accept.

I recommend that the following minor corrections be considered:

1. It seems that something is wrong in the calculation of the number of cases in cohort A in fig 1.

106217-23661-3094-32-702-321=78407

Please confirm this.

2. Still, in my computer environment, the column of CCI in Table 1 seems to be broken; please confirm this.

7. PLOS authors have the option to publish the peer review history of their article (what does this mean?). If published, this will include your full peer review and any attached files.

Reviewer #1: **Yes: **Savvas Lampridis

Reviewer #2: No

---

## [Author Response · Author response to Decision Letter 1]

14 May 2024

Response to reviewers

We greatly appreciate the opportunity to revise our manuscript based on the insightful and constructive comments from the reviewers. We are hopeful that the revised manuscript will be considered suitable for publication, thereby making valuable contributions to the field and the journal's readership. We look forward to the potential publication of our work in PLOS ONE and are available to provide any further information or clarifications as needed.

Editor

Additional Editor Comments – Thank you for your lots of efforts to improve article. Please confirm comments from Reviewer #2, then reply for the 2 questions.

Authors’ response:

Thank you for your time and effort in reviewing our manuscript. We are pleased to meet the standards set by the editor and reviewers. The revision process was a true pleasure for all authors, and we hope our study benefits the academic community.

Reviewer #1

Comment - Thank you for the attention and consideration you have shown to my suggested revisions for your manuscript. It is evident that a significant amount of effort and thought has been directed towards the refining of your work, integrating the feedback provided during the peer review process. The resulting modifications demonstrate a thorough approach and significantly improve the rigor and overall quality of your manuscript. I look forward to witnessing the impact your research will undoubtedly have on the academic community.

Authors’ response:

We sincerely appreciate your time and effort in reviewing our manuscript. Based on your comments and suggestions, we have significantly enhanced the quality of our manuscript and incorporated essential points. Thank you for your valuable feedbacks.

Reviewer #2

Comment - I would like to thank the authors for their considerable effort regarding the revision. My previous comments were appropriately addressed. Therefore, my decision is provisional accept.

Authors’ response:

We truly appreciate your time and effort in reviewing our manuscript and are pleased to meet the journal's standards. The revision process was a great experience for all authors. Thank you.

Comment #1 – It seems that something is wrong in the calculation of the number of cases in cohort A in fig 1. 106217-23661-3094-32-702-321=78407 Please confirm this. 

Authors’ response:

Thank you for pointing this out. We have confirmed that the first number in the flow chart for the pre-COVID-19 cohort was incorrectly entered as the total number of claims instead of the total number of patients. This has been corrected, and the remaining values have been verified for accuracy. We apologize for any confusion caused.

Comment #2 – Still, in my computer environment, the column of CCI in Table 1 seems to be broken; please confirm this.

Authors’ response – We have re-checked Table 1 on all authors' PCs but could not identify the issue with the broken view. We suggest the editorial office check if there is any problem with Table 1 before publication. We have attached a PDF version of Table 1 for the reviewer to verify the contents. We apologize for any inconvenience. Thank you.

---

## [Decision Letter · Decision Letter 2]

24 May 2024

Trends in degenerative lumbar spinal surgery during the early COVID-19 pandemic in Republic of Korea: a national study utilizing the National Health Insurance Database

PONE-D-23-41698R2

Dear Dr. Kim,

We’re pleased to inform you that your manuscript has been judged scientifically suitable for publication and will be formally accepted for publication once it meets all outstanding technical requirements.

Kind regards,

Kentaro Yamada, M.D., Ph.D.

Academic Editor

PLOS ONE

Reviewers' comments:

Reviewer #1: Thank you for your continued efforts in addressing Reviewer #2’s concern regarding the calculation of the number of cases in cohort A, as depicted in Figure 1. Regarding Reviewer #2’s second comment, I did not find any issues with Table 1 formatting either.

Congratulations on your successful revisions and the forthcoming publication of your work in PLoS One.

Reviewer #2: Thanks for the prompt reply. I have ensured that the authors have appropriately addressed the revisions.

---

## [Editor Report · Acceptance letter]

31 May 2024

PONE-D-23-41698R2 

PLOS ONE

Dear Dr. Kim, 

I'm pleased to inform you that your manuscript has been deemed suitable for publication in PLOS ONE. Congratulations! Your manuscript is now being handed over to our production team.

Kind regards, 

on behalf of

Dr. Kentaro Yamada 

Academic Editor

PLOS ONE